# DrivingWorld: Constructing World Model for Autonomous Driving via Video GPT

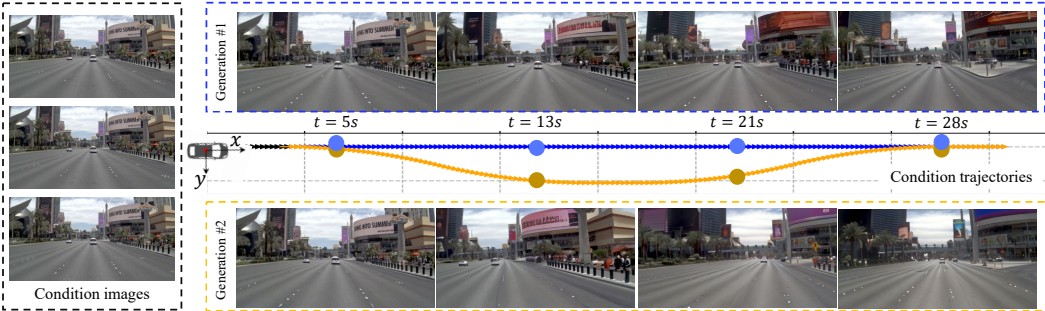

Figure 1: **An example of controllable generation**. We design two trajectories for the same scene to generate different driving scenarios, *i.e.*, one moving straight forward and the other with a curved path.

## Abstract

Recent successes in autoregressive (AR) generation models, such as the GPT series in natural language processing, have motivated efforts to replicate this success in visual tasks. Some research aims to extend this approach to autonomous driving by building video-based world models capable of generating realistic future video sequences and predicting the ego states. However, the prior works tend to produce unsatisfactory results, since the classic GPT framework is designed to handle 1D contextual information, such as text, and lacks the inherent capability to model the spatial and temporal dynamics necessary for video generation. In this paper, we present *DrivingWorld*, a GPT-style world model for autonomous driving with several spatial-temporal fusion mechanisms. This design allows for effective modeling of both spatial and temporal dynamics, enabling high-fidelity, long time video generation. Specifically, we first propose next-state-prediction strategy to model temporal coherence between consecutive frames and then apply next-token-prediction strategy to capture spatial information within a frame. To further enhance generalization ability, we propose a novel masking strategy and reweight strategy for token prediction to mitigate long time drifting issues and enable precise control. Our work is capable of producing high-fidelity and consistent video clips with long-time duration. Experiments demonstrate that, in contrast to prior works, our method delivers superior visual quality and significantly more accurate controllable future video generation. Visit our project page at `https://anonymous.4open.science/r/DrivingWorld-5714`.

## 1 Introduction

In recent years, autoregressive (AR) learning schemes have achieved significant success in natural language processing, as demonstrated by models like the GPT series (Radford, 2018; Radford et al., 2019; Brown, 2020). These models predict future text responses from past data, making AR approaches as leading candidates in the pursuit of Artificial General Intelligence (AGI). Inspired by these advancements, many researchers have sought to replicate this success in visual tasks, such as building vision-based world models for autonomous driving (Hu et al., 2023).

A critical capability in autonomous driving systems is future event prediction (Guan et al., 2024). However, many prediction models rely heavily on large volumes of labeled data, making them vulnerable to out-of-distribution and long-tail scenarios (Santana & Hotz, 2016; Wang et al., 2023; Lu et al., 2023). This is especially problematic for rare and extreme cases, such as accidents, where obtaining sufficient training data is challenging. A promising solution lies in autoregressive world models, which learn comprehensive information from unlabeled data like massive videos through unsupervised learning. This enables more robust decision-making in driving scenarios. These world models have the potential to reason under uncertainty and reduce catastrophic errors, thereby improving the generalization and safety of autonomous driving systems.

The prior work, GAIA-1 (Hu et al., 2023), was the first to extend the GPT framework from language to video, aiming to develop a video-based world model. Similar to natural language processing, GAIA transforms 4D temporally correlated frames into a sequence of 1D feature tokens and employs the next-token prediction strategy to generate future video clips. However, the classic GPT framework, primarily designed for handling 1D contextual information, *lacks the inherent capability to effectively model the spatial and temporal dynamics necessary for video generation*. As a result, the videos produced by GAIA-1 often suffer from low quality and noticeable artifacts, highlighting the challenge of achieving high fidelity and coherence within a GPT-style video generation framework.

In this paper, we introduce *DrivingWorld*, a driving world model built on a GPT-style video generation framework. Our primary goal is to enhance the modeling of temporal coherence in an autoregressive framework to create more accurate and reliable world models. To achieve this, our model incorporates three key innovations: 1) **Temporal-Aware Tokenization**: We propose a temporal-aware tokenizer that transforms video frames into temporally coherent tokens, reformulating the task of future video prediction as predicting future tokens in the sequence. 2) **Hybrid Token Prediction**: Instead of relying solely on the next-token prediction strategy, we introduce a next-state prediction strategy to better model temporal coherence between consecutive states. Afterward, the next-token prediction strategy is applied to capture spatial information within each state. 3) **Long-time Controllable Strategies**: To improve robustness, we implement random token dropout and balanced attention strategies during autoregressive training, enabling the generation of longer-duration videos with more precise control.

Overall, our work enhances temporal coherence in video generation using the AR framework, learning meaningful representations of future evolution. Experiments show that the proposed model achieves good generalization performance, is capable of generating long-duration video sequences, and provides accurate next-step trajectory predictions, maintaining a reasonable level of controllability.

## 2 RELATED WORK

**World Model.** The world model (LeCun, 2022) captures a comprehensive representation of the environment and forecasts future states based on a sequence of actions. World models has been extensively explored in both game (Hafner et al., 2019; 2020; 2023) and laboratory environments (Wu et al., 2023). Dreamer (Hafner et al., 2019) trains a latent dynamics model using past experiences to forecast state values and actions within a latent space. DreamerV2 (Hafner et al., 2020) builds upon the original Dreamer model, reaching human-level performance in Atari games. DreamerV3 Hafner et al. (2023) employs larger networks and successfully learns to acquire diamonds in Minecraft from scratch. DayDreamer (Wu et al., 2023) extends Dreamer (Hafner et al., 2019) to train four robots directly in the real world, successfully tackling locomotion and manipulation tasks. Recently, world models for driving scenarios have garnered significant attention in both academia and industry. Most previous works (Argenson & Dulac-Arnold, 2020; Diehl et al., 2021; 2023; Henaff et al., 2019) have been limited to simulators or well-controlled lab environments. Drive-WM (Wang et al., 2024b) explores real-world driving planners using diffusion models. GAIA-1 (Hu et al., 2023) investigates real-world driving planners based on autoregressive models, but GAIA-1 has large parameters and computational demands, which increase as the number of condition frames grows. In this paper, we propose an efficient world model in an autoregressive framework for autonomous driving scenarios.

**VQVAE.** VQVAE (Van Den Oord et al., 2017) learns a discrete codebook representation via vector quantization to model image distributions. VQGAN (Esser et al., 2021) improves realism by incorporating LPIPS loss (Zhang et al., 2018) and adversarial PatchGAN loss (Isola et al., 2017). MoVQ (Zheng et al., 2022) tackles VQGAN's spatially conditional normalization issue by embedding

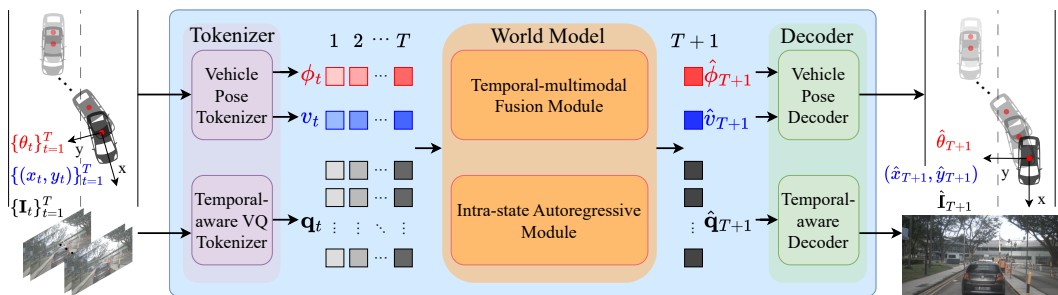

Figure 2: **Pipeline of *DrivingWorld***. The vehicle orientations $\{\theta_t\}_{t=1}^T$, ego locations $\{(x_t, y_t)\}_{t=1}^T$, and an image sequence $\{\mathbf{I}_t\}_{t=1}^T$ are taken as the conditional input, which are first tokenized as latent embeddings. Then our proposed multi-modal world model attempts to comprehend them and forecast the future states, which are detokenized to the vehicle orientation $\hat{\theta}_{T+1}$, location $(\hat{x}_{T+1}, \hat{y}_{T+1})$, and the image $\hat{\mathbf{I}}_{T+1}$. With the autoregressive process, we can generate over 30 seconds videos.

spatially variant information into quantized vectors. LlamaGen (Sun et al., 2024) further fine-tunes VQGAN, showing that a smaller codebook vector dimension and a larger codebook size enhance reconstruction performance. While VQGAN-based structures are widely used, some methods explore more efficient architectures. ViT-VQGAN (Yu et al., 2021) replaces the convolutional encoder-decoder with a Vision Transformer (ViT), improving the model's ability to capture long-range dependencies. VAR (Tian et al., 2024) employs a multi-scale structure to predict subsequent scales based on previous ones, enhancing both generation quality and speed. However, these methods focus on single-image processing, making it unable to capture temporal consistency. To address this, we propose a temporal-aware tokenizer and decoder in this paper.

**Autoregressive Image Generation.** Early autoregressive models for image generation operate at the pixel level, using CNNs (Van den Oord et al., 2016) or RNNs (Van Den Oord et al., 2016) to predict unknown pixels in a zigzag manner. Inspired by language models and enabled by VQVAE (Van Den Oord et al., 2017), images are encoded into discrete tokens. This allows autoregressive models to shift from operating in pixel space to working in a discrete token space. VQGAN (Esser et al., 2021), DALL-E (Ramesh et al., 2021), and Parti (Yu et al., 2022a) introduce VQVAE to encode continuous images to discrete tokens, and use autoregressive models to generate images in a way of next token prediction. VAR (Tian et al., 2024) decomposes an image into residuals at multiple resolutions, encoding each resolution's residuals into a different number of discrete tokens. During the next scale prediction, all tokens at a specific resolution are predicted at once, reducing the number of autoregressive steps. Llamagen (Sun et al., 2024) demonstrates that vanilla autoregressive models, which use the same "next-token prediction" approach as language models, can also achieve state-of-the-art performance in image generation. In this paper, we take it a step further by applying autoregressive models to video generation.

**Video Generation.** Currently, there are three mainstream video generation models: GAN-based, diffusion-based, and GPT-based methods. GAN-based methods (Yu et al., 2022b; Skorokhodov et al., 2022; Tian et al., 2021) often face several challenges, such as mode collapse, where the diversity of the videos generated by the generator becomes limited. Additionally, the adversarial learning between the generator and discriminator can lead to instability during training. One major issue with diffusion-based methods is their inability to generate precisely controlled videos because the stochastic nature of the diffusion process introduces randomness at each step, making it difficult to enforce strict control over specific attributes in the generated content. On the other hand, traditional GPT-based methods (Yan et al., 2021; Han et al., 2022) allow for a certain level of control, but their computational cost grows quadratically with the sequence length, significantly impacting model efficiency. In this paper, we propose a decoupled spatio-temporal world model framework, which ensures precise control while significantly reducing computational cost and improving model efficiency.

## 3 METHOD

Our proposed world model, *DrivingWorld*, leverages a GPT-style architecture to predict future states with high efficiency, capable of extending predictions beyond 30 seconds at a frequency of 6Hz. This

model is designed to comprehend past real-world states and forecast future video content and vehicle motions. *DrivingWorld* is specifically focused on predicting the next frame status at time $T + 1$ based on the historical states from time 1 to $T$, and we can generate long videos by sequentially predicting future states one by one. Each state at time $t$ is represented as $[\theta_t, (x_t, y_t), \mathbf{I}_t]$, where $\theta_t$ is the vehicle's orientation, $(x_t, y_t)$ is its location, and $\mathbf{I}_t$ is the current image. Figure 2 presents the pipeline. Our proposed *DrivingWorld* not only generates future states $[\theta_{T+1}, (x_{T+1}, y_{T+1}), \mathbf{I}_{T+1}]$ based on past observations $\{[\theta_t, (x_t, y_t), \mathbf{I}_t]\}_{t=1}^T$, but also supports controllable simulation of complex driving scenarios by manipulating the vehicle's location and orientation. Section 3.1 details our proposed tokenizers for encoding temporal multimodal information into the unified latent space. To model the relationships between long time sequential states, we introduce a GPT-style temporal multimodel decoupled world model in Section 3.2. To extract the state from the tokens predicted by the world model, we also introduce a temporal decoder, which is discussed in detail in Section 3.3. Additionally, we introduce long-time controllable strategies in Section 3.4 to address the drift problem and enhance the robustness of the proposed world model.

## 3.1 TOKENIZER

Tokenization (Zheng et al., 2022; Van Den Oord et al., 2017) converts continuous data into discrete tokens, enabling integration with language models and enhanced multimodal sequence modeling. In our approach, the tokenizer maps multimodal states into a unified discrete space, which enables accurate and controllable multimodal generation. To produce temporally consistent embeddings for images, we propose a temporal-aware vector quantized tokenizer. Our proposed vehicle pose tokenizer discretizes pose trajectories and integrates them into our *DrivingWorld*.

**Prelimilary: Single Image Vector Quantized Tokenizer.** The single image vector quantized (VQ) tokenizer, as described in Van Den Oord et al. (2017), is designed to convert an image feature map $\mathbf{f} \in \mathbb{R}^{H \times W \times C}$ to discrete tokens $\mathbf{q} \in [K]^{H \times W}$. The quantizer utilizes a learned discrete codebook $\mathcal{Z} \in \mathbb{R}^{K \times C}$, containing $K$ vectors, to map each feature $\mathbf{f}^{(i,j)}$ to the index $\mathbf{q}^{(i,j)}$ of the closest code in $\mathcal{Z}$. This method enables the conversion of continuous image data into discrete tokens.

**Temporal-aware Vector Quantized Tokenizer.** Single image VQ tokenizers often struggle to produce temporally consistent embeddings, leading to discontinuous video predictions and hindering the training of the world model. The image sequence $\{\mathbf{I}_t\}_{t=1}^T$ is encoded as $\{\mathbf{f}_t\}_{t=1}^T$, where each feature is processed independently and thus lacks temporal information.

To address this issue, we propose a temporal-aware vector quantized tokenizer designed to ensure consistent embeddings over time. Specifically, to capture temporal dependencies, we insert a self-attention layer both *before* and *after* VQGAN quantization, where the attention operates along the temporal dimension. This allows our model to capture long-range temporal relationships between frames, improving coherence and consistency in the generated sequences. Our model builds upon the open-source VQGAN implementation from LlammaGen (Sun et al., 2024). The integration of our straightforward yet effective temporal self-attentions can be seamlessly incorporated into the original framework, followed by fine-tuning to develop a robust and generalizable temporal-aware VQ tokenizer. $\{\mathbf{f}_t\}_{t=1}^T$ are fed into temporal self-attention $\mathcal{H}(\cdot)$ before performing quantization, thus the tokens are:

$$\mathbf{q}_t^{(i,j)} = \underset{k \in [K]}{\arg\min} \left\| \text{lookup}(\mathcal{Z}, k) - \mathcal{H}(\mathbf{f}_1^{(i,j)}, ..., \mathbf{f}_T^{(i,j)})[t] \right\|_2, \quad \mathbf{q}_t^{(i,j)} \in [K], \tag{1}$$

where $\text{lookup}(\mathcal{Z}, k)$ means taking the $k$-th vector in codebook $\mathcal{Z}$.

**Vehicle Pose Tokenizer.** To accurately represent the vehicle's ego status, including its orientation $\theta$ and location $(x, y)$, we adopt a coordinate system centered at the ego vehicle, as depicted in Figure 2. Instead of global poses, we adopt the relative poses between adjacent frames. This is because that global poses present a significant challenge due to the increasing magnitude of absolute pose values over long-term sequences, making normalization difficult and reducing model robustness. As the model generates longer sequences, these large pose values become harder to manage, ultimately complicating the handling of long-term video generation.

For the sequence of the vehicle's orientation $\{\theta_t\}_{t=1}^T$ and location $\{(x_t, y_t)\}_{t=1}^T$, we propose to compute relative values for each time step with respect to the previous one. The relative location and orientation at the first time step is initialized as zero. The ego-centric status sequence is given

by $\{\Delta\theta_t\}_{t=1}^T$ and $\{(\Delta x_t, \Delta y_t)\}_{t=1}^T$. To tokenize them, we discretize the ego's surrounding space. Specifically, we discretize the orientation into $\alpha$ categories, and the $X$ and $Y$ axes into $\beta$ and $\gamma$ bins, respectively. Thus, the relative trajectory at time $t$ is tokenized as follows:

$$\phi_t = \left\lfloor \frac{\Delta\theta_t - \theta_{min}}{\theta_{max} - \theta_{min}}\alpha \right\rfloor, \quad v_t = \left\lfloor \frac{\Delta x_t - x_{min}}{x_{max} - x_{min}}\beta \right\rfloor \cdot \gamma + \left\lfloor \frac{\Delta y_t - y_{min}}{y_{max} - y_{min}}\gamma \right\rfloor. \tag{2}$$

Finally, we process the past $T$ real-world states $\{[\theta_t, (x_t, y_t), \mathbf{I}_t]\}_{t=1}^T$ and tokenize them into a discrete sequence $\{[\phi_t, v_t, \mathbf{q}_t]\}_{t=1}^T$, where each token is a discrete representation of the vehicle's state at each time step.

## 3.2 World Model

The world model aims to comprehend past inputs, mimic real-world dynamics, and predict future states. In our context, it forecasts upcoming driving scenarios and plans a feasible future trajectory. To do this, the world model concatenates historical state tokens $\{[\phi_t, v_t, \mathbf{q}_t]\}_{t=1}^T$ into a long sequence, where the 2D image tokens are unfolded into a 1D form in zig-zag order. Thus the objective is to predict next-state status $\mathbf{r}_{T+1} = (\phi_{T+1}, v_{T+1}, \mathbf{q}_{T+1}^1, \ldots, \mathbf{q}_{T+1}^{H \times W})$ based on the sequence of past observations $\{\mathbf{r}_t\}_{t=1}^T$, capturing both temporal and multimodal dependencies. Note that all discrete tokens from different modalities are mapped into a shared latent space by their respective learnable embedding layers before being fed to the world model, i.e. $\mathbf{h}_t = Emb(\mathbf{r}_t)$. All subsequent processes are conducted within this latent space.

**Prelimilary: Next-Token Prediction.** A straightforward method is to use the GPT-2 (Radford et al., 2019) structure for 1-D sequential next-token prediction. Figure 3 (a) shows a simplified example. The causal attention is applied for long-text prediction and the $i$-th token in $T + 1$ is modeled as:

$$\hat{\mathbf{r}}_{T+1}^i = \mathcal{G}([sos], \mathbf{r}_1, \ldots, \mathbf{r}_T, \hat{\mathbf{r}}_{T+1}^1, \ldots, \hat{\mathbf{r}}_{T+1}^{i-1}) \tag{3}$$

where $[sos]$ denotes the start-of-sequence token, $\mathbf{r}$ is the ground truth tokens, $\hat{\mathbf{r}}$ is the predict tokens, and $\mathcal{G}$ represent GPT-2 (Radford et al., 2019) model. However, such a 1-D design is inadequate for our specific scenarios. Predicting long videos requires generating tens of thousands of tokens, which is significantly time-consuming. Additionally, it overlooks the spatially structured image features inherent in images.

Therefore, we propose a next-state prediction pipeline, which consists of two modules: one integrates temporal and multimodal information for next-state feature generation (i.e. *Temporal-multimodal Fusion Module*), and the other is an autoregressive module (i.e. *Intra-state Autoregressive Module*) for high-quality intra-state token generation.

**Temporal-multimodal Fusion Module.** Our temporal-multimodal module is composed of a separate temporal layer and a multimodal layer. This decouples the processing of temporal and multimodal information, thereby improving both training and inference speed while also reducing GPU memory consumption. As shown in the Figure 3, We propose to employ a causal attention mask in the temporal transformer layer $\mathcal{F}_a(\cdot)$, where each token only attends to itself and tokens at the same sequential position from all previous frames. This is designed to fully leverage temporal information.

$$\tilde{\mathbf{h}}_t^i = \mathcal{F}_a(\mathbf{h}_1^i, \ldots, \mathbf{h}_t^i), \ i \in [1, H \times W + 2], \ t \in [1, T]. \tag{4}$$

In the multimodal information fusion layer $\mathcal{F}_b(\cdot)$, we employ a bidirectional mask in the same frame, which is designed to fully integrate intra-state multimodal information and facilitates interactions between modalities. Each token attends to other tokens from the same time step,

$$\mathring{\mathbf{h}}_t = \mathcal{F}_b(\tilde{\mathbf{h}}_t), \ t \in [1, T]. \tag{5}$$

The temporal and multimodal layers are alternately stacked for $N$ layers to form this module.

**Intra-state Autoregressive Module.** After the temporal-multimodal module, we obtain features for future frame state prediction. A naive approach is to predict next-state tokens $h_t$ at the same time. Recently, multiple image-generation works (Sun et al., 2024) propose that an autoregressive pipeline for next-token prediction generates better images, and even outperforms diffusion methods. Inspired by this, we propose an intra-state autoregressive module for next-time action and image generation,

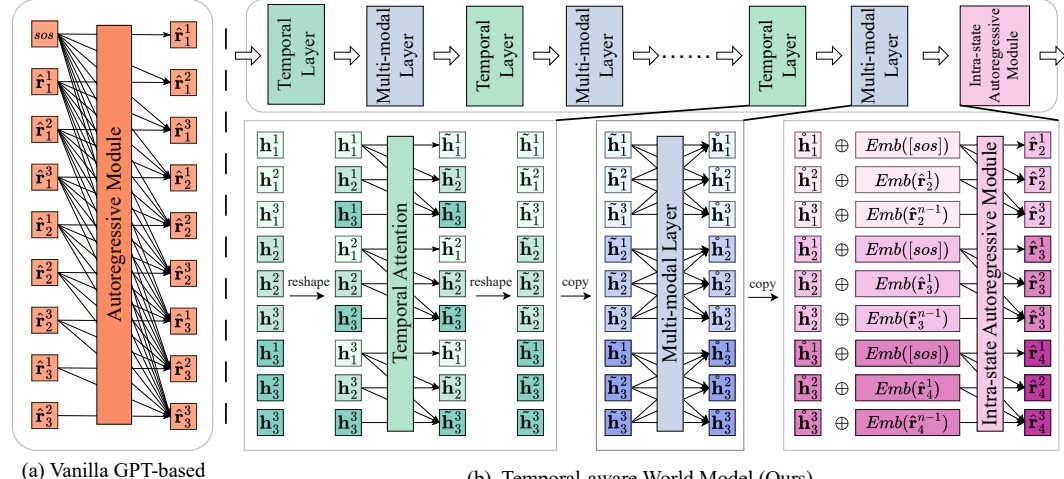

(a) Vanilla GPT-based World Model

(b) Temporal-aware World Model (Ours)

Figure 3: **Inference illustration of Vanilla GPT and temporal-aware GPT (ours).** For simplicity, we assume a video clip only has 3 frames and each frame consists of only 3 tokens, where $x_2^1$ denotes $1^{st}$ token of the $2^{nd}$ video frame. (a) The vanilla GPT places all tokens in a 1D sequence and employ the causal attention, which can autoregressively predicts next tokens. (b) We propose a temporal-multimodal fusion module to meld multi-modal information $\{\mathbf{h}_t^i\}_{i=1}^n$ and obtain the next-state feature $\{\mathring{\mathbf{h}}_{t+1}^i\}_{i=1}^n$. To generate high-quality next-state videos and vehicle tokens, we employ the causal attention, thus such tokens ($\{\hat{\mathbf{r}}_{t+1}^i\}_{i=1}^n$) are autoregressively predicted. $Emb(\cdot)$ denotes the embedding of corresponding tokens. In the temporal layer, each token only attends to itself and tokens at the same sequential position from all previous frames. The multi-modal layer and intra-state autoregressive module are separately operated to the tokens per frame.

see Figure 3. Specifically, to predict $\hat{\mathbf{r}}_{T+1} = (\hat{\mathbf{r}}_{T+1}^1, \ldots, \hat{\mathbf{r}}_{T+1}^{H \times W+2})$, we add the temporal-multimodal output feature $\mathring{\mathbf{h}}_T = (\mathring{\mathbf{h}}_T^1, \ldots, \mathring{\mathbf{h}}_T^{H \times W+2})$ with the sequential tokens ($[sos], \hat{\mathbf{r}}_{T+1}^1, \ldots, \hat{\mathbf{r}}_{T+1}^{H \times W+1}$). Then they are input to the intra-state autoregressive transformer layers $\mathcal{F}_c(\cdot)$. The causal mask is employed in these layers, thus each token can only attend itself and prefix intra-state tokens. The autoregressive process is present in Eq. 6. As our pipeline incorporates both the next-state prediction and the next intra-state token prediction, we enforce two teacher-forcing strategies in training, i.e. one for the frame level and the other one for the intra-state level.

$$\hat{\mathbf{r}}_{T+1}^i = \mathcal{G}(Emb([sos]) + \mathring{\mathbf{h}}_T^1, Emb(\hat{\mathbf{r}}_{T+1}^1) + \mathring{\mathbf{h}}_T^1, \ldots, Emb(\hat{\mathbf{r}})_{T+1}^{i-1} + \mathring{\mathbf{h}}_T^i), \ i \in [1, H \times W+2]. \quad (6)$$

We use cross-entropy loss for training, as

$$\mathcal{L}_{WorldModel} = -\sum_{t=1}^{T+1} \sum_{j=1}^{H \times W+2} \log P(\hat{\mathbf{r}}_t^j | \mathbf{r}_{<t}, \mathbf{r}_t^1, \ldots, \mathbf{r}_t^{j-1}). \quad (7)$$

where $\mathbf{r}$ is the ground truth tokens, and $\hat{\mathbf{r}}$ is the predict tokens.

### 3.3 DECODER

By predicting the next-state tokens $\hat{\mathbf{r}}_{T+1} = (\hat{\phi}_{T+1}, \hat{v}_{T+1}, \hat{\mathbf{q}}_{T+1})$ using the world model, we can then leverage the decoder to generate the corresponding relative orientation $\Delta\hat{\theta}_{T+1}$, relative location $(\Delta\hat{x}_{T+1}, \Delta\hat{y}_{T+1})$, and the reconstructed image $\hat{\mathbf{I}}_{T+1}$ for that state. This process allows us to map the predicted latent representations back into physical outputs, including both spatial and visual data.

**Vehicle Pose Decoder.** For the predicted relative orientation token $\hat{\phi}_{T+1}$ and relative location token $\hat{v}_{T+1}$, we can obtain the corresponding values through the inverse function of the Eq. 2 as follows:

$$\Delta\theta_t = \theta_{min} + \frac{\phi_t}{\alpha}(\theta_{max} - \theta_{min}),$$

$$\Delta x_t = x_{min} + \frac{1}{\beta}\left\lfloor\frac{v_t}{\gamma}\right\rfloor(x_{max} - x_{min}), \Delta y_t = y_{min} + \left(\frac{v_t}{\gamma}v_t - \left\lfloor\frac{v_t}{\gamma}\right\rfloor\right)(y_{max} - y_{min}). \quad (8)$$

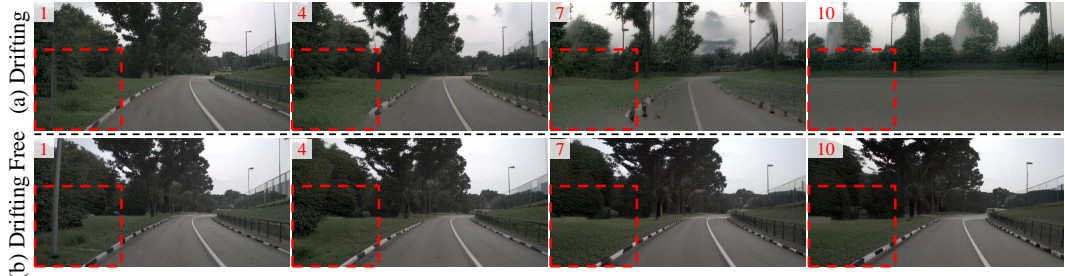

Figure 4: **Example of autoregressive drift.** Without our proposed masking strategy in training, when generating only 10 frames, the scene will quickly corrupt.

**Temporal-aware Decoder.** For the predicted image tokens $\hat{\mathbf{q}}_{T+1}$, we retrieve the corresponding feature from the codebook $\mathcal{Z} \in \mathbb{R}^{K \times C}$ in the Temporal-aware Vector Quantized Tokenizer. Note that after the quantization layer we insert a temporal self-attention to enhance the temporal consistency.

$$\hat{\mathbf{f}}_{T+1}^{i,j} = \text{lookup}(\mathcal{Z}, \hat{\mathbf{q}}_{T+1}^{i,j}), \; i \in [1, H], \; j \in [1, W]. \tag{9}$$

### 3.4 LONG-TIME CONTROLLABLE GENERATION

**Token Dropout for Drifting-free Autoregression.** During training, the world model uses past ground-truth tokens as conditioning to predict the next tokens. However, during inference, the model must rely on previously generated tokens for conditioning, which may contain imperfections. Training solely with perfect ground-truth images can lead to a content drifting problem during inference, causing rapid degradation and eventual failure in the generated outputs. To address this, we propose a random masking strategy (RMS), where some tokens from ground-truth tokens are randomly dropped out. Each token has a 50% chance of being replaced by another random token in this frame, and this dropout is applied to the entire conditioning image sequence with a probability of 30%. As shown in Figure 4, this dropout strategy significantly mitigates the drifting issue during inference.

**Balanced Attention For Precise Control.** The world model utilizes extensive attention operations to exchange and fuse information among tokens. However, each video frame in our experiments contains 512 tokens, while only 2 tokens represent the ego pose (orientation and location). This imbalance can cause the model to overlook pose signals, leading to unsatisfactory controllable generation. To address this, we propose a balanced attention operation to achieve more precise control by prioritizing ego state tokens in the attention mechanism, rather than attending to all tokens equally. Specifically, we manually increase the weights of the orientation and location tokens in the attention map (before the `softmax` layer), adding constant weights of 0.4 and 0.2, respectively, to these tokens. Additionally, we incorporate QK-norm (Henry et al., 2020) and 2D-rotary positional encoding (Su et al., 2024) to further stabilize training and enhance performance.

## 4 EXPERIMENTS

### 4.1 IMPLEMENTATION DETAILS

**Tokenizer and Decoder.** The video tokenizer has 70M trainable parameters. The images are with size of $256 \times 512$ and tokenized into 512 tokens. Considering 2 tokens for orientation and location respectively, each state consists of $514$ tokens. The size of adopted codebook is set to $16,384$. The model is trained for 1,000k steps with a total batch size of 128 distributed across 32 NVIDIA 4090 GPUs. The traing images are selected from Openimages (Kuznetsova et al., 2020), COCO (Lin et al., 2014), YoutubeDV (Zhang et al., 2022), and NuPlan (Caesar et al., 2021) datasets. We train the temporal-aware VQ tokenizer and decoder using a combination of three loss functions: charbonnier loss (Lai et al., 2018), perceptual loss from LPIPS (Zhang et al., 2018), and codebook loss (Van Den Oord et al., 2017) (see Apendix for more details).

**World Model.** The world model module has 1 billion parameters and is trained on video sequences consisting of 16 frames. The first 15 frames serve as conditional inputs, with the final frame used for

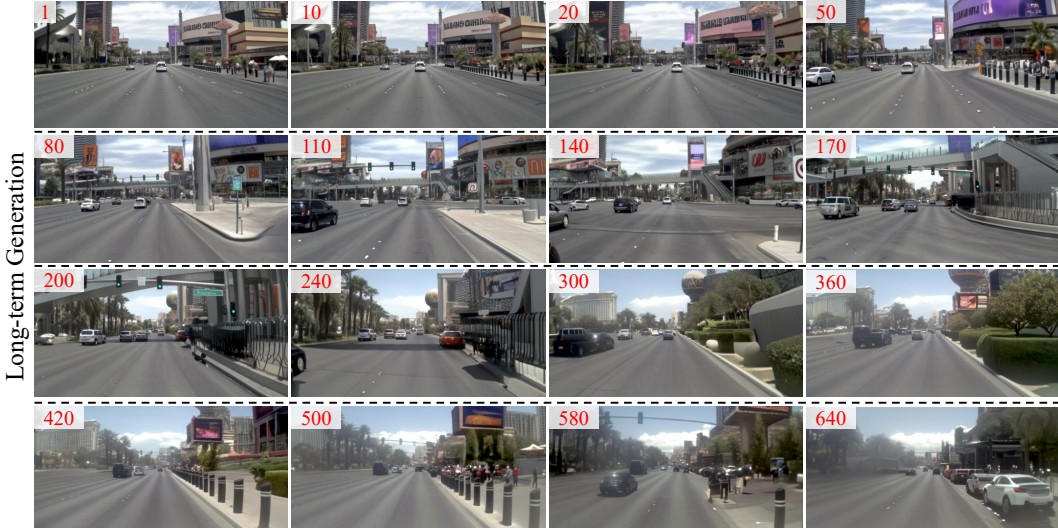

Figure 5: **Long time video generation.** We present some videos generated by our method, each with 640 frames at 5Hz, i.e. 128 seconds. Please notice the coherent 3D scene structures captured by our method across different frames (please see the digital version).

supervision. All video clips are sampled at 5 Hz. With 514 tokens per image, the sequence consists of a total of 7,710 tokens. Training is conducted over 9 days, spanning 300k iterations with a fixed learning rate of $1 \times 10^{-4}$ and a batch size of 64, distributed across 64 NVIDIA A100 80GB GPUs. The model is trained on the NuPlan (Caesar et al., 2021) dataset, which comprises 120 hours of human driving data downsampled to 5 Hz. We segment each scene into 75-frame clips, discarding any that fall short of this length, resulting in a training set of 28,000 clips.

**Evaluation Dataset and Metrics.** We use 200 video clips from the NuPlan (Caesar et al., 2021) test dataset as our test set, with each clip containing 100 frames sampled at 5 Hz. Additionally, we include 150 video clips from the NuScenes (Caesar et al., 2020) test set as part of our evaluation. The quality of video generation is assessed using the Frechet Video Distance (FVD), and we also report the Frechet Inception Distance (FID) to evaluate image generation quality.

### 4.2 LONG TIME VIDEO GENERATION

One of the key advantages of our method is its ability to generate long-duration videos. As shown in Figure 5, we visualize one long-duration video generated by our model. By conditioning on just 15 frames, our model can generate up to 640 future frames at 5 Hz, resulting in 128-second videos with strong temporal consistency. These results demonstrate that our model maintains high video fidelity and preserves 3D structural integrity across the generated frames. In contrast, previous methods often struggle with drifting or degradation in long-duration videos. The ability to generate extended video sequences underscores our model's potential for tasks that require long-term predictions, such as autonomous driving or video synthesis in complex dynamic environments.

### 4.3 COMPARISON WITH OTHER METHODS

**Quantitative Comparison of Generated Videos.** We provide the quantitative comparison with several methods on the NuScenes (Caesar et al., 2020) dataset in Table 1. Since most methods are not publicly available, we use the results reported in their respective papers for comparison. Our model achieves competitive performance with a FID of 16.4 and an FVD of 174.4, despite being trained exclusively on the NuPlan dataset (Caesar et al., 2021). Although Drive-WM and GenAN report slightly better FID and FVD scores, they were both trained on the NuScenes dataset (Caesar et al., 2020). Moreover, our method is capable of generating significantly longer videos than them.

**Qualitative Comparison of Generated Videos.** We provide a qualitative comparison with Stable Video Diffusion (Blattmann et al., 2023), a well-known public video diffusion model, on the Nu-

Table 1: **Generation comparison on the NuScenes validation set.**

| Metric | DriveGAN (Santana & Hotz, 2016) | DriveDreamer (Wang et al., 2023) | WoVoGen (Lu et al., 2023) | Drive-WM (Wang et al., 2024a) | GenAD (Yang et al., 2024) | *DrivingWorld* Ours |
|---|---|---|---|---|---|---|
| FID ↓ | 73.4 | 52.6 | 27.6 | 15.8 | 15.4 | 16.4 |
| FVD ↓ | 502.3 | 452.0 | 417.7 | 122.7 | 184.0 | 174.4 |
| Generation Length (s) | N/A | 4 | 2.5 | 8 | 4 | 30 |
| Zero-shot | ✗ | ✗ | ✗ | ✗ | ✗ | ✓ |

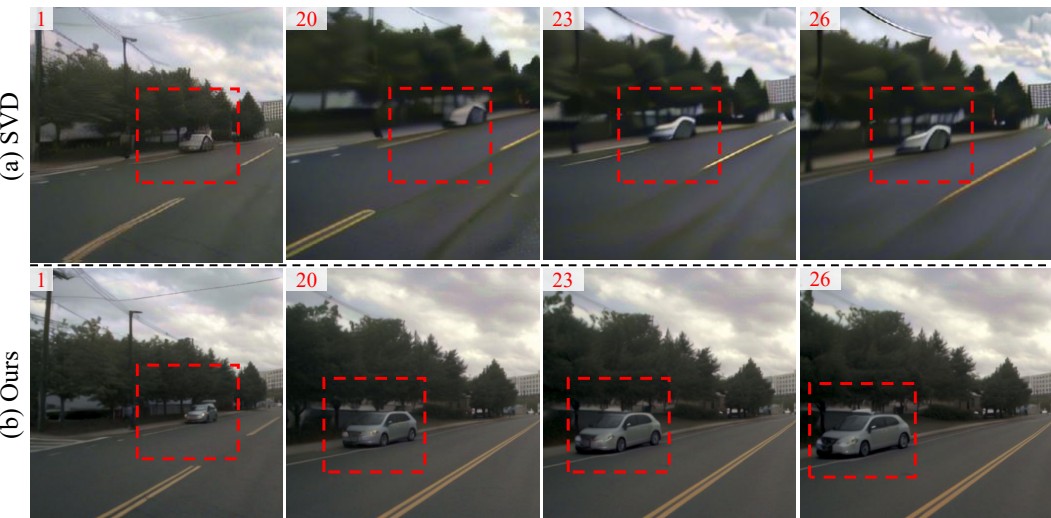

Figure 6: **Comparison of SVD and ours.** We compare our method with SVD for generating 26 frames on a zero-shot NuScenes scene. In these moderately long-term videos, our method better preserves street lane details and maintains car identity more effectively.

Plan (Caesar et al., 2021) dataset. As shown in Figure 6, our generated videos demonstrate superior temporal consistency, particularly in maintaining details like street lanes and vehicles.

**Quantitative Comparison of Image Tokenizers.** We further evaluate our temporal-aware image tokenizer against those proposed in other works. Since the image tokenizer is part of a VQVAE, we assess the encoding-decoding performance of these VQVAEs instead. The experiments, conducted on the NuPlan (Caesar et al., 2021) dataset, are summarized in Table 2. The VQVAE models from VAR (Tian et al., 2024) and VQGAN (Esser et al., 2021) demonstrate reasonable image quality in terms of PSNR and LPIPS scores, but both fall short on FID and FVD metrics. In contrast, Llama-Gen's VQVAE (Sun et al., 2024) shows significant improvements in FID and FVD scores. After fine-tuning it on driving scenes, we observe further enhancements in FVD performance. Ultimately, our temporal-aware VQVAE outperforms all others, enhancing temporal consistency and achieving the best scores across all four metrics.

Table 2: **Quantitative comparison of different VQVAE methods.** The evaluations are performed on the $256 \times 512$ NuPlan datasets.

| VQVAE Methods | $FVD_{12}$ ↓ | FID ↓ | PSNR ↑ | LPIPS ↓ |
|---|---|---|---|---|
| VAR | 164.66 | 11.75 | 22.35 | 0.2018 |
| VQGAN | 156.58 | 8.46 | 21.52 | 0.2602 |
| Llama-Gen | 57.78 | 5.99 | 22.31 | 0.2054 |
| Llama-Gen Finetuned | 20.33 | 5.19 | 22.71 | 0.1909 |
| Temporal-aware (Ours) | **14.66** | **4.29** | **23.82** | **0.1828** |

## 4.4 ABLATION STUDY

**Setting.** Due to the prolonged training time and computational costs, we experiment on a smaller dataset for the ablation study. We extract 12 hours of video data from the NuPlan (Caesar et al., 2021)

dataset for training, and select 20 videos from NuPlan (Caesar et al., 2021) test sets to create our testing data. All ablation experiments are conducted on 32 NVIDIA A100 80GB GPUs with a total batch size of 3. Each model is trained from scratch for 50,000 iterations, requiring approximately 96 GPU hours.

**Model Structure *w/* and *w/o* Random Masking Strategy.** To evaluate the impact of our random masking strategy on model robustness, we experiment model training with and without random token masking. This masking process simulates potential prediction errors during inference, enhancing the model's ability to handle noise. As shown in Table 3, the model trained without masking experiences a significant performance decline on NuPlan (Caesar et al., 2021) dataset, particularly in long term videos where inference errors are more prevalent as we can see from the $FVD_{40}$ scores. Generally speaking, disabling masking results in a substantial increase in FVD, with a rise of 4 to 32 percent across different scenarios, indicating poor generalization and reduced robustness against noisy inputs.

Table 3: **Comparison of *w/* and *w/o* Random Masking Strategy.** Removing the random masking strategy during training ("*w/o* Masking") leads to drifting, resulting in degraded performance on NuPlan dataset.

| Methods | $FVD_{10} \downarrow$ | $FVD_{25} \downarrow$ | $FVD_{40} \downarrow$ |
|---|---|---|---|
| *w/o* Masking | 449.40 | 595.49 | 662.60 |
| Ours | **445.22** | **574.57** | **637.60** |

Table 4: **Performance comparison between our method and GPT-2.** Our method not only improves efficiency but also produces better results on NuPlan dataset.

| Methods | $FVD_{10} \downarrow$ | $FVD_{25} \downarrow$ | $FVD_{40} \downarrow$ |
|---|---|---|---|
| GPT-2 | 2976.97 | 3505.22 | 4017.15 |
| Ours | **445.22** | **574.57** | **637.60** |

Table 5: **Memory usage (GB) analysis of our method and vanilla GPT.** Our method consumes much lower GPU memory than vanilla GPT structure.

| Num. of frames | 5 | 6 | 7 | 8 | 9 | 10 | 15 |
|---|---|---|---|---|---|---|---|
| vanilla GPT | 31.555 | 39.305 | 47.237 | 55.604 | 66.169 | 77.559 | OOM |
| Ours | 21.927 | 24.815 | 26.987 | 29.877 | 31.219 | 34.325 | 45.873 |

**Discussion With Vanilla GPT structure.** We compare the memory usage of our *DrivingWorld* structure with the vanilla GPT architecture, specifically GPT-2, which processes tokens sequentially across all frames during inference. GPT-2's serial token prediction slows down performance, significantly increasing computational burden and memory usage. As shown in Table 5, GPT-2's memory consumption grows quadratically with sequence length, making it inefficient for long sequences. In contrast, our method separates temporal and multimodal dependencies, allowing for more efficient representation and computation. As sequence lengths increase, our model maintains stable computational costs and memory usage, avoiding the sharp scaling seen in GPT-2. Moreover, our approach not only enhances efficiency but also improves result quality. As shown in Table 4, our model outperforms GPT-2 in FVD scores on NuPlan (Caesar et al., 2021) dataset.

## 5 CONCLUSION AND FUTURE WORK

In conclusion, *DrivingWorld* addressed the limitations of previous video generation models in autonomous driving by leveraging a GPT-style framework to produce longer, high-fidelity video predictions with improved generalization. Unlike traditional methods that struggle with coherence in long sequences or rely heavily on labeled data, *DrivingWorld* generated realistic, structured video sequences while enabling precise action control. Compared to the classic GPT structure, our proposed spatial-temporal GPT structure adopts next-state-prediction strategy to model temporal coherence between consecutive frames and then apply next-token-prediction strategy to capture spatial information within a frame. Looking ahead, we plan to incorporate more multimodal information and integrate multiple visual inputs. By fusing data from various modalities and viewpoints, we aim to improve action control and video generation accuracy, enhancing the model's ability to understand complex driving environments and further boosting the overall performance and reliability of autonomous driving systems.

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

# A APPENDIX

We present more experiments in the appendix and provide additional video results on the website (Click here to see more videos).

## A.1 DIFFERENT CONDITION FRAMES

To investigate the effect of varying the number of condition frames on model performance, we conduct a series of experiments by gradually increasing the length of the condition frames used during training and inference. We extract 12 hours of video data from the NuPlan dataset for training, and select 20 videos from each of the NuScenes and NuPlan test sets to create our testing data, referred to as NuScenes-Small and NuPlan-Small, respectively. As shown in Table 6, the model consistently improves as the number of condition frames increases. Specifically, when fewer condition frames are used, the model struggles to capture long-term dependencies. In contrast, with longer condition frames, the model has more temporal context to work with, allowing it to better understand the environment and generate more precise outputs.

Table 6: **Ablation the amount of condition frames.** DrivingWorld generates better videos (lower FVD error) when conditioning more frames.

| Number of | Nuscene-Small | | | Nuplan-Small | | |
| Condition Frames | $FVD_{10} \downarrow$ | $FVD_{25} \downarrow$ | $FVD_{40} \downarrow$ | $FVD_{10} \downarrow$ | $FVD_{25} \downarrow$ | $FVD_{40} \downarrow$ |
|---|---|---|---|---|---|---|
| 5 | 475.14 | 802.35 | 1113.81 | 494.86 | 597.95 | 679.05 |
| 10 | 448.93 | 719.57 | 965.62 | 449.29 | 577.54 | 646.60 |
| 15 | 440.27 | 695.26 | 933.13 | 445.22 | 574.57 | 637.60 |
| 25 | 360.55 | 546.11 | 721.56 | 400.94 | 512.73 | 580.10 |

## A.2 THE EFFECT OF INTRA-STATE AUTOREGRESSIVE MODULE

To assess the impact of the final intra-state autoregressive (AR) module on our *DrivingWorld*'s overall performance, we perform an ablation study by removing this module from the model structure. Thus future state's tokens are predicted simultaneously. The experimental results, as summarized in Table 7, indicate that the absence of the AR module leads to a noticeable decrease in performance across FVD metric. Note that 'Baseline-w/o AR' and 'Ours' have comparable model size. Specifically, removing the AR module results in an increase from 18% to 71% in FVD metric, which suggests that the module plays a crucial role in capturing sequential dependencies and refining the final output predictions in the long-term generation.

Table 7: **Ablation the effect of intra-state autoregressive module.** 'Baseline-w/o AR' removes the intra-state autoregressive module and generates all next-state tokens simultaneously, while 'Ours' autoregressively generates next-state tokens, which have much lower FVD error.

| Methods | Nuscene-Small | | | Nuplan-Small | | |
| | $FVD_{10} \downarrow$ | $FVD_{25} \downarrow$ | $FVD_{40} \downarrow$ | $FVD_{10} \downarrow$ | $FVD_{25} \downarrow$ | $FVD_{40} \downarrow$ |
|---|---|---|---|---|---|---|
| Baseline-w/o AR | 523.53 | 1052.30 | 1601.36 | 525.04 | 729.75 | 1007.91 |
| Ours | 440.27 | 695.26 | 933.13 | 445.22 | 574.57 | 637.60 |

## A.3 SCALING LAW OF OUR *DrivingWorld*.

To investigate the scaling law of our model, we conducted a series of ablation experiments by progressively scaling up the number of parameters in the model. As shown in Table 8, increasing the model size consistently leads to improved performance. In smaller models, the limited capacity hinders the ability to fully capture the complexity of the data, resulting in suboptimal performance, especially on long-term generation.

## A.4 MODEL STRUCTURE *w/* AND *w/o* RANDOM MASKING STRATEGY

To evaluate the impact of our random masking strategy on model robustness, we experiment model training with and without random token masking. This masking process simulates potential prediction

Table 8: **Experiment of scaling law of our model.** We compare three different model sizes (i.e. 10M, 100M, 1B). Larger model can generate much better videos on Nuscense and Nuplan-small datasets.

| Methods | Nuscene-Small | | | Nuplan-Small | | |
|---|---|---|---|---|---|---|
| | $FVD_{10} \downarrow$ | $FVD_{25} \downarrow$ | $FVD_{40} \downarrow$ | $FVD_{10} \downarrow$ | $FVD_{25} \downarrow$ | $FVD_{40} \downarrow$ |
| 10M | 654.95 | 1248.53 | 1817.82 | 816.39 | 1003.03 | 1262.31 |
| 100M | 463.72 | 809.02 | 1120.30 | 481.25 | 609.20 | 915.01 |
| 1B | 440.27 | 695.26 | 933.13 | 445.22 | 574.57 | 637.60 |

errors during inference, enhancing the model's ability to handle noise. As shown in Table 9, the model trained without masking experiences a significant performance decline on Nuscene dataset, particularly in long term videos where inference errors are more prevalent as we can see from the $FVD_{40}$ scores.

## A.5 DISCUSSION WITH VANILLA GPT STRUCTURE

We compare the performance of our *DrivingWorld* structure with the vanilla GPT architecture, specifically GPT-2, which processes tokens sequentially across all frames during inference. As shown in Table 10, our model outperforms GPT-2 in FVD scores on Nuscene dataset.

Table 9: **Effect of Random Masking Strategy.** Removing the random masking strategy during training ("*w/o* Masking") leads to drifting, resulting in degraded performance on Nuscene dataset.

| Methods | $FVD_{10} \downarrow$ | $FVD_{25} \downarrow$ | $FVD_{40} \downarrow$ |
|---|---|---|---|
| *w/o* Masking | 450.04 | 711.01 | 965.16 |
| Ours | **440.27** | **695.26** | **933.13** |

Table 10: **Performance comparison between our method and GPT-2.** Our method not only improves efficiency but also produces better results on Nuscense dataset.

| Methods | $FVD_{10} \downarrow$ | $FVD_{25} \downarrow$ | $FVD_{40} \downarrow$ |
|---|---|---|---|
| GPT-2 | 2122.24 | 3200.29 | 4063.26 |
| Ours | **440.27** | **695.26** | **933.13** |

## A.6 LONG-TERM GENERATION

To demonstrate the long-time generation ability of our *DrivingWorld*, we create more long-time generation videos, which are enclosed in the supplementary materials. Condition frames are marked with white borders, while generated frames are in red borders.

