# OpenReview forum: "DrivingWorld: Constructing World Model for Autonomous Driving via Video GPT"
_ICLR.cc/2025/Conference — ICLR 2025 Conference Withdrawn Submission_

### Official Review · Reviewer_1FhW · 2024-11-03

**Soundness:** 3
**Presentation:** 3
**Contribution:** 3
**Rating:** 5
**Confidence:** 5

**Summary:**

In this paper, a GPT-style model called DrivingWorld is introduced. DrivingWorld incorporates several spatial-temporal fusion mechanisms to effectively model both spatial and temporal dynamics, enabling high-fidelity, long-duration video generation. Experiments confirm that this proposed method is capable of producing high-fidelity and consistent video clips over extended periods.

**Strengths:**

1. The proposed temporal-aware Image Tokenizer achieves the highest scores across all four metrics, indicating robust performance.
2. The paper reflects a significant effort and comprehensive workload.

**Weaknesses:**

1. There appears to be an error in line 488 regarding the batch size; it might be beneficial to verify whether it should be 32 instead of 3.
2. The experiments indicate that the RMS strategy does not perform optimally. It might be beneficial to assess its effectiveness under a full experimental setting.
3. The experiments are conducted solely on nuPlan or nuScenes datasets. The generalization capability of the 1B model across different datasets needs further evaluation to ensure it does not overfit to the nuPlan series dataset.
4. The concept of the World Model may be considered controversial and could benefit from further clarification to establish its acceptance and validity within the field.

**Questions:**

1. Could you provide more details on the implementation specifics of training on 32 NVIDIA 4090 GPUs?
2. What is the resolution of the methods reported in Table 1? Also, could you explain why longer videos result in a slightly higher FVD (e.g., 122.7 vs 174.4)?
3. Could you elaborate on the computational costs associated with integrating self-attention into the Temporal-aware Image Tokenizer?
4. How does the performance of DrivingWorld compare to that of the Llama series?

---

### Official Review · Reviewer_vk6g · 2024-11-03

**Soundness:** 1
**Presentation:** 1
**Contribution:** 1
**Rating:** 1
**Confidence:** 1

**Summary:**

The paper breaks the author's anonymity. I could find the full author list with their affiliations and personal web pages in two clicks, i.e., first click on the link in the abstract of the paper to go to the repo, then click on the "index.html" there.
This violates the statement given by the authors "Anonymous Url: I certify that there is no URL (e.g., github page) that could be used to find authors’ identity." thus I believe the paper has to be withdrawn.

Here is the snippet of the index.html file containing the paper title and personal information of the authors:

```
             <h1 class="title is-1 publication-title"> <a style="color:#D46EE8">DrivingWorld</a>:  Constructing World Model for Autonomous Driving via Video GPT</h1>
            <!-- <div class="is-size-5 publication-authors"> -->
              <!-- Paper authors -->
              <!-- <span class="author-block">
                <a href="https://huxiaotaostasy.github.io/" target="_blank">Xiaotao Hu<sup>1,2,*</sup></a>,</span>
                <span class="author-block"> -->

                <!-- <a href="https://yvanyin.net/" target="_blank">Wei Yin<sup>2,*</sup></a>,</span>
                <span class="author-block">

                <a href="https://scholar.google.com/citations?user=fcpTdvcAAAAJ&hl=en&oi=ao" target="_blank">Mingkai Jia<sup>1,2</sup></a>,</span>
                <span class="author-block">

                <a href="#" target="_blank">Junyuan Deng<sup>1,2</sup></a>,</span>
                <span class="author-block">

                <a href="https://scholar.google.com/citations?user=CrK4w4UAAAAJ&hl=en&oi=ao" target="_blank">Xiaoyang Guo<sup>2</sup></a>,</span>
                <span class="author-block">

                <a href="https://scholar.google.com/citations?hl=en&user=pCY-bikAAAAJ" target="_blank">Qian Zhang<sup>2</sup></a>,</span>
                <span class="author-block">

                <a href="https://www.xxlong.site/" target="_blank">Xiaoxiao Long<sup>1,†</sup></a>,</span>
                <span class="author-block">

                <a href="https://scholar.google.com/citations?user=XhyKVFMAAAAJ&hl=en&oi=ao" target="_blank">Ping Tan<sup>1</sup></a></span>
                <span class="author-block">

                </div> -->

                  <!-- <div class="is-size-5 publication-authors">
                    <span class="author-block">
                      <sup>1</sup> Hong Kong University of Science and Technology |  <sup>2</sup>Horizon Robotics <br>
                    </span>
...
```

**Strengths:**

-

**Weaknesses:**

-

**Questions:**

-

**Details Of Ethics Concerns:**

-

---

### Official Review · Reviewer_ceym · 2024-11-04

**Soundness:** 3
**Presentation:** 3
**Contribution:** 3
**Rating:** 6
**Confidence:** 5

**Summary:**

This paper proposes DrivingWorld, a self-driving world model based on the GPT architecture for generating high-fidelity and long-term video sequence predictions. The model improves performance through three key innovations: Temporal-Aware Tokenization, Hybrid Token Prediction, and Long-time Controllable Strategies. Experiments show that the model can generate more than 100 seconds of high-quality video at a frequency of 5Hz. Compared with the traditional GPT structure, this method significantly reduces the computational cost by decoupling spatiotemporal processing while maintaining better temporal consistency and structural integrit

**Strengths:**

1. The paper demonstrates clarity in its presentation and organization. The writing flows logically from motivation to implementation, while the figures illustrate key concepts and results.
2. The paper's core innovation, applying temporal-aware GPT architecture to autonomous driving world modeling, represents an advancement in the field.
3. The introduction of Dropout for Drifting-free Autoregression is a good solution to generating long-driving video sequences. Experiments show this technique addresses the common problem of quality degradation in long-term predictions.

**Weaknesses:**

1. The biggest concern lies in the model's controllability claims. While the paper demonstrates models of "controllability" with vehicle trajectory control, this is primarily restricted to lane changes on straight roads with minimal viewpoint variations. The surrounding environment, including other vehicles and road structures, remains purely autoregressive without direct control.
Besides, the absence of more challenging scenarios like turning left/right in intersections raises questions about the model's true generalization capabilities and control flexibility.
2. The generated videos still exhibit noticeable visual artifacts and physical inconsistencies. For instance, the black car in Figure 5's "300s" subfigure and the project webpage's Example 3 demonstrate unrealistic vehicle collision scenarios at the 14-second mark.
These issues highlight a considerable challenge: the model's inability to handle physically complex scenarios where vehicles are in close proximity or potential collision situations, which are crucial for autonomous driving applications.
3. The method's long video generation is excellent, but the model's FID (16.4) and FVD (174.4) scores, while reasonable, do not lead the benchmark comparisons in Table 1. This becomes more apparent when considering recent SOTA works like DiVE and Vista, which are absent from the comparison.

**Questions:**

The paper contains inconsistent  descriptions regarding the video generation frequency, stating 6Hz in the method section ( line 161 "capable of extending predictions beyond 30 seconds at a frequency of 6Hz. " ) but 5Hz in the experiment section (line 416 "our model can generate up to 640 future frames at 5 Hz, resulting in 128-second videos with strong temporal consistency.")

---

### Official Review · Reviewer_P3Lz · 2024-11-04

**Soundness:** 3
**Presentation:** 3
**Contribution:** 3
**Rating:** 6
**Confidence:** 3

**Summary:**

The paper presents an approach to constructing a world model for autonomous driving using a GPT-style architecture, which is an application of autoregressive models in the visual domain. The authors claim that their model, DrivingWorld, is capable of high-fidelity, long-duration video generation with improved temporal coherence and controllability. The experiments and comparisons with existing methods are well-documented, and the paper is generally well-organized and clearly written.

**Strengths:**

1. The idea of extending the GPT framework to video generation for autonomous driving is innovative and has the potential to significantly impact the field.
2. The proposed spatial-temporal fusion mechanisms and the next-state-prediction strategy are well-thought-out and seem technically sound.
3. The paper provides a thorough set of experiments demonstrating the model's capabilities, including comparisons with other state-of-the-art methods.
4. The paper is well-structured, and easy to read.

**Weaknesses:**

1. Temporal-aware Vector Quantized Tokenizer has been extensively utilized in video generation fields, such as SORA, and thus does not constitute a technological contribution.
2. It is not entirely clear how well the model generalizes to diverse driving scenarios beyond those seen in the training data. Additional experiments or analysis on how the model performs on out-of-distribution data would strengthen the paper.
3. The paper could benefit from a discussion on the computational resources required for training and inference, especially considering the model's size. Discussing potential optimizations or trade-offs in computational efficiency could make the paper more appealing to practitioners.
4. The paper missed some relevant works, such as "VDT: General-purpose Video Diffusion Transformers via Mask Modeling" (ICLR2024).

**Questions:**

Please see weaknesses.

---

### Note · Authors · 2024-11-13

I have read and agree with the venue's withdrawal policy on behalf of myself and my co-authors.